# No evidence that late-sighted individuals rely more on color for object recognition: A Bayesian generalized mixed effects model analysis

*Thomas S. A. Wallis[1,2]\* and Joshua M. Martin[1]*

**1** Centre for Cognitive Science and Institute for Psychology, Technical University of Darmstadt.
**2** Center for Mind, Brain and Behavior (CMBB), Universities of Marburg, Giessen and Darmstadt.

\*Corresponding author. Email: thomas.wallis@tu-darmstadt.de

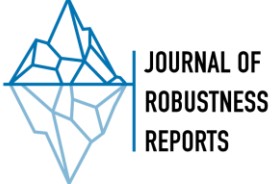

## Abstract

**A Bayesian mixed-effects analysis found no evidence that patients treated with cataract surgery rely more on color cues for object recognition compared to controls.**

## Target article

M. Vogelsang et al., Impact of early visual experience on later usage of color cues. *Science, 384*, 907-912 (2024). doi: 10.1126/science.adk9587

## 1 Goal

The aim of this analysis was to investigate whether a Bayesian generalised mixed effects model using binomial error distributions would support the claim that individuals treated for congenital blindness via cataract removal surgery (Prakash patients) rely more on color cues for object recognition than age-matched controls [1]. The original conclusion was based on the finding that the difference in proportion correct between color and greyscale images was smaller for controls (Figure 1A): patients improved more when color was available. The statistical evidence for this finding was based on a set of t-tests of the difference scores. However, because accuracy is bounded and some control participants perform at or near ceiling, t-tests on difference scores may underestimate the improvement in controls by treating a 5% change near 95% as statistically equivalent to one at 50% (i.e. assuming that probability is linear and error is Normal). We therefore examined whether this effect is robust when analyzing trial-level accuracy using a logistic regression model [2], which represents performance changes on the unbounded log-odds scale (Figure 1B).

## 2      Methods

We fit Bayesian generalized linear mixed-effects models, assuming a binomial outcome distribution (correct/incorrect) with a logistic link function. To assess robustness to prior assumptions, we tested four weakly informative priors based on previous recommendations [3,43]: normal distributions with standard deviations of 10, 5, and 1, and a Cauchy distribution with scale 2.5. The fixed effects included image type (color vs. grayscale), group (Prakash vs. control), and their interaction (binary variables were coded with [-0.5, 0.5]). To account for repeated measures, we specified random intercepts and slopes for image type as part of a maximal random effects structure justified by the experimental design [5].

We evaluated the hypothesized interaction effect (i.e., greater color benefit in the Prakash group) by comparing models with and without the interaction term using Bayes factors and by examining the posterior distributions of the interaction coefficients. The analysis was performed using the brms package [6] in R (see code for full analysis).

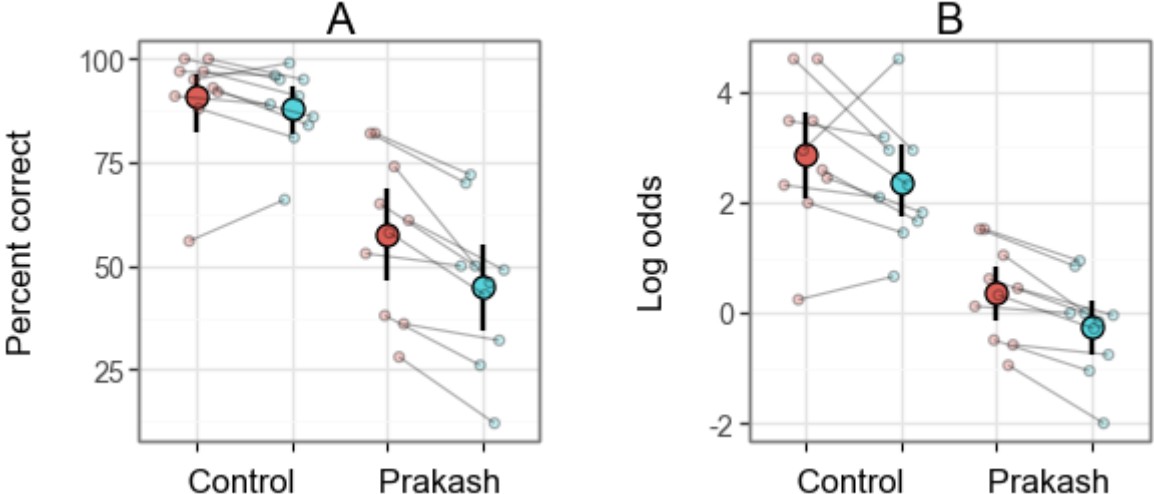

*Figure 1: Comparison of recognition performance modelled on the percent correct scale (Panel A) versus the log−odds (logit) scale (Panel B). Each point represents a participant's performance in the color (red) and grayscale (blue) conditions; large points and error bars indicate group means ± standard error. For visualization purposes, two values of 100% were converted to 99% to avoid infinite values. While the percent correct scale (A) is intuitive, it compresses changes near the performance bounds (0% and 100%), potentially underestimating differences.*

## 3      Results

Posterior mean estimates for the interaction effect regression coefficient ranged from $-0.093$ to $-0.115$ across priors, with 95% credible intervals spanning from $[-0.646, 0.315]$ to $[-0.574, 0.492]$, all of which included zero. Bayes factors ranged from 0.030 to 0.321, consistently favoring the additive model without interaction, indicating moderate to very strong evidence [7] against the presence of an interaction effect.

Model diagnostics indicated reliable estimation: all $\hat{R}$ values were between 1.00-1.01, effective sample sizes for key parameters exceeded recommended thresholds, and posterior

predictive checks showed close alignment between observed and simulated data (see code for details).

# 4    Conclusion

Vogelsang et al. [1] reported a significant group difference using t-tests, suggesting that Prakash patients improved more than controls when color was available. In contrast, our Bayesian mixed-effects analysis found evidence in favor of a model with no interaction between group and image type, indicating that Prakash patients were, if anything, equally impacted by the removal of color information in an image recognition task. Expressing performance changes as raw percentages (assuming a linear probability model) results in the color and greyscale conditions in the control group being compressed closer together (Figure 1A). This compression directly creates the interaction effect observed using t-tests. We argue that assuming a linear probability model with Normal measurement error is therefore inappropriate here. Using a generative model that treats this performance data on its natural scale (log odds and binomial errors) mitigates the effects of performance saturation by attributing greater weight to changes at or near ceiling. Therefore, the conclusion that late-sighted individuals rely more on color information than individuals with normal visual development from these data hinges on which modeling approach one views as most appropriate for the underlying data-generating process.

## Acknowledgments and Disclosures

**Reproducibility**      We were able to computationally reproduce the original analysis and results.

**Code and Data Availability**    Our analysis code can be found at a hosted repository at the following link:
https://github.com/ag-perception-wallis-lab/bayesian_reanalysis_vogelsang2024

**Author Contributions**        Thomas Wallis and Joshua Martin contributed equally to all roles in this work. The order of authorship was determined by a coin toss.

**Funding**        Funded by the European Union (ERC, SEGMENT, 101086774). Views and opinions expressed are however those of the author(s) only and do not necessarily reflect those of the European Union or the European Research Council. Neither the European Union nor the granting authority can be held responsible for them. We thank Pawan Sinha and Lukas Vogelsang for their comments on an earlier version of our work, and for making their dataset available.

**Conflicts of Interest**   The authors declare no conflicts of interest.

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
