# Peer review of "No evidence that late-sighted individuals rely more on color for object recognition: A Bayesian generalized mixed effects model analysis"

_Journal of Robustness Reports_

## Round 1 · Referee Report · Zoltan Kekecs (Referee 1) · 2025-7-9

Strengths

1-reproducible analysis
2-open data, easy to understand open code
3-breif focused report

Weaknesses

1-the rationale for the different analysis approach is unclear
2-it is unclear what cases the difference in the conclusions between the two analytical approaches

Report

This is a short focused paper. I like the approach used by tha authors. I was able to analytically reproduce their results, and to reproduce a similar result with a frequentist alternative of the model. I think the report is worthy of publication, after the authors clarify it a little bit, why they suggest using this alternative analysis approach over the original one, and relatedly, what could be the reason for the different result and conclusion compared to that by the origial researchers.

Requested changes

1-the atuhors should clarify why they suggest using this alternative analysis approach over the original one 2- the atuhors should clarify what is the main reason for why this new analysis results in opposite conclusion compared to the original paper.

Recommendation

Ask for minor revision

---

## Round 1 · Referee Report · Henrik Singmann (Referee 2) · 2025-9-10

Strengths

  1. Reproducible analysis in R markdown format.
  2. Clear and short paper.
  3. Nice figure.

Weaknesses

  1. It is unclear why the Bayesian mixed-effects analysis should be preferred to the one reported in the original paper.
  2. Code for Figure 1 in report not included in provided R code.
  3. The conclusion talks about two different criticisms: The differences in results between t-test and Bayesian-GLMM and learning as a "competing causal explanation". However, the paper only focuses on the differences in analysis so the second issue does not follow from the presented analysis.

Report

I think this is an interesting paper that provides an important addition to the Vogelsang et al. (2024) paper (which appeared in Science and was already cited 14 times). It also provides a clear and reasonable alternative analysis approach (GLMM) and shows that with this alternative approach a different results pattern occurs.

I was also able to reproduce their analysis and confirmed the results myself using a frequentist analysis which shows the same pattern of results (no interaction effect when using a logistic-binomial GLMM, but an interaction effect when using either repeated-measures ANOVA or a linear mixed-effects model assuming a normal response distribution; see: https://gist.github.com/singmann/cd196db4b796c38cfd14160c37c83e9c).

However, the paper currently does not do a good enough job arguing why the alternative analysis approach should be preferred to the original author's approach. What actually is wrong with the original author's approach (t-test)?

To be transparent, I do not think that using a binomial-logistic GLMM is necessarily better in this situation than a paired t-test as done by the original authors. Which analysis approach one ultimately prefers depends on which assumptions one thinks is more believable. For the binomial GLMM, one assumes that participants are normally distributed around their condition means in logistic space. For the t-test, one essentially assumes that participants are normally distributed around their condition means in accuracy space. Which assumption is correct is not an easy question.

Recently, I am becoming more convinced by analyses that do not require a data transformation (such as to logistic space) for accuracy data. One reason for this is a series of papers (e.g., Gomilla, 2021; Jaccard & Brinberg, 2021) arguing for the usage of models on a non-transformed probability space. In this context, models such as paired t-test or RM-ANOVA for accuracy data is also known as the linear probability model.

Taken together, in a revision the authors need to make absolutely clear that the evidence for the interaction hinges on whether one is willing to believe the linear probability model (as done by Vogelsang et al., 2024) or the binomial GLMM with logistic link function. In other words, there is some ambiguity towards the evidence for the interaction. Which side one falls on depends on which statistical model one believes provides a better account of the data generating process.

References

  • Gomila, R. (2021). Logistic or linear? Estimating causal effects of experimental treatments on binary outcomes using regression analysis. Journal of Experimental Psychology: General, 150(4), 700–709. https://doi.org/10.1037/xge0000920
  • Jaccard, J., & Brinberg, M. (2021). Monte Carlo simulations using extant data to mimic populations: Applications to the modified linear probability model and logistic regression. Psychological Methods, 26(4), 450–465. https://doi.org/10.1037/met0000383

Requested changes

  1. Add code for Figure 1 to code repository.
  2. Rewrite conclusion and clearly separate the discussion of statistical issues from issues regarding the experimental design (learning as a possible alternative explanation for the pattern).
  3. Provide a more balanced discussion of the statistical differences between both analysis approaches. Why do the authors think a binomial GLMM is more justifiable here than the linear probability model employed by Vogelsang et al. (2024).

Recommendation

Ask for minor revision

---

## Round 2 · Referee Report · Henrik Singmann (Referee 2) · 2025-10-30

Report
The revision addresses all of my previous criticisms. It now provides a balanced perspective on the evidence provided by the new analysis. This manuscript is ready to be published as is.
I have only two small comments to make, neither of which should affect publication of the manuscript as is.
- I believe the following comment made by the authors in their response to my previous review is not correct. In particular, the authors misconstrue the difference between Bayesian and frequentist GLMMs. Contrary to their claims, both frequentist and Bayesian GLMMs are essentially the exact same model with the same assumption regarding the group-level distribution (i.e., the individual effects follow a normal distribution), namely $p(y)=\int p(y\vert u)\,p(u)\,du$ (where $y$ is the response and $u$ the random effect parameters). The main technical difference between both models is that in the frequentist case the integral needs to be solved as part of the likelihood equation whereas it is solved through MCMC integration in the Bayesian case. While Bayesian models also include priors, this is not really material to the distribution of the individual-level estimates. Both types of models imply pretty much the exact same amount of group-level shrinkage on the random effect terms. To be more concrete, frequentist models due not impose a more "rigid" group-level structure. In both cases the actual distribution of the group-levels can be non normal if indicated by the data (e.g., Schielzeth et al., 2020). As this is not part of the manuscript nothing needs to be done.
In a Bayesian model estimated via MCMC, assigning a Normal prior to random effects does not force their posterior distributions to be Normal: if the data suggest skew or other deviations, the posterior will reflect that. This is because each participant-level mean effect has its own posterior distribution (i.e. is a parameter to be estimated). This differs from frequentist implementations (e.g., using lme4 with REML), where the Normal random-effects assumption is rigidly imposed through the estimation of the variance of a Normal distribution, with each participant-level offset being then determined by the BLUP “Best Linear Unbiased Predictors” (see e.g. Baayen et al., 2008).
- I appreciate that the code for Figure 1 was added. But I am a bit disappointed it is in Python and not in R, given that the rest of the code is in R. This somehow reduces the computational reproducibility of the manuscript. I do not think this should prevent publication, but it would be better if it were also in R.
Reference
Schielzeth, H., Dingemanse, N. J., Nakagawa, S., Westneat, D. F., Allegue, H., Teplitsky, C., Réale, D., Dochtermann, N. A., Garamszegi, L. Z., & Araya‐Ajoy, Y. G. (2020). Robustness of linear mixed‐effects models to violations of distributional assumptions. Methods in Ecology and Evolution, 11(9), 1141–1152. https://doi.org/10.1111/2041-210X.13434
Recommendation
Publish (meets expectations and criteria for this Journal)

---

## Round 2 · Referee Report · Zoltan Kekecs (Referee 1) · 2025-10-30

Report
Recommendation
Publish (easily meets expectations and criteria for this Journal; among top 50%)

---

## Round 2 · Author Response

List of changes
The main point of both reviewers relates to the rationale for our approach and what could explain the divergent findings of the two methods. Reviewer 1 writes that “The rationale for the different analysis approach is unclear” and “The authors should clarify what is the main reason for why this new analysis results in opposite conclusion compared to the original paper” while reviewer 2 writes that “It is unclear why the Bayesian mixed-effects analysis should be preferred to the one reported in the original paper” and relatedly “What actually is wrong with the original author’s approach (t-test)?”
Response:
We agree that this could have been made clearer in the original manuscript. Our original motivation for the reanalysis derived from our observation of a ceiling effect in the control data, where many data-points are compressed together (Figure 1A). T-tests on difference scores underestimate these changes by treating percentage differences identically across the performance bound. A t-test implicitly assumes a linear model with Normal-distributed errors, which as a generative model for accuracy data is impossible (since it can predict performance greater than 100%). A logistic regression model, on the other hand, represents performance changes on an unbounded log-odds scale, which can mitigate this issue by being more sensitive to changes occurring near ceiling. Additionally, logistic regression typically assumes a binomial error distribution, which is appropriate for data in which the unit of observation is a binary outcome. We have now added the following to the introduction to make the rationale for this approach clearer:
“However, because accuracy is bounded and some control participants perform at or near ceiling, t-tests on difference scores may underestimate the improvement in controls by treating a 5% change near 95% as statistically equivalent to one at 50% (i.e. assuming that probability is linear and variance is Normally-distributed). We therefore examined whether this effect is robust when analyzing trial-level accuracy using a logistic regression model [2], which represents performance changes on the unbounded log-odds scale (Figure 1B).”
Furthermore, we believe that the observed differences in conclusions arise from this fundamental distinction between how the two methods treat performance changes on bounded versus unbounded scales. We have added this to the conclusion as an interpretation of the divergent findings:
“Expressing performance changes as raw percentages (assuming a linear probability model) results in the color and greyscale conditions in the control group being compressed closer together (Figure 1A). This compression directly creates the interaction effect observed using t-tests. We argue that assuming a linear probability model with Normal measurement error is therefore inappropriate here. Using a generative model that treats this performance data on its natural scale (log odds and binomial errors) mitigates the effects of performance saturation by attributing greater weight to changes at or near ceiling.”
Four smaller points raised by reviewer 2
In addition to the main point above, there are four smaller points that we would like to address in turn. We list the quote relevant to each point and our brief response underneath.
1. “Taken together, in a revision the authors need to make absolutely clear that the evidence for the interaction hinges on whether one is willing to believe the linear probability model (as done by Vogelsang et al., 2024) or the binomial GLMM with logistic link function.”
We agree that ultimately the conclusion that one draws from the study hinges on which model one thinks is more faithful to the underlying generative process. We have added the following as a concluding sentence to reflect this:
"Therefore, the conclusion that late-sighted individuals rely more on color information than individuals with normal visual development from these data appears to hinge on which modeling approach one views as most appropriate for the underlying data-generating process."
2. “Code for Figure 1 in report not included in provided R code.”
We thank the reviewer for pointing this out. We have now updated the repository to include the code for generating the figure to make sure every aspect of the report is transparent and reproducible.
3. “The conclusion talks about two different criticisms: The differences in results between t-test and Bayesian-GLMM and learning as a "competing causal explanation". However, the paper only focuses on the differences in analysis so the second issue does not follow from the presented analysis.”
We agree with the reviewer that this statement does not directly derive from our alternative statistical analysis. While we feel it is an important design limitation related to the experiment, we have decided to remove this point as it is not directly relevant to the report at hand.
4. “To be transparent, I do not think that using a binomial-logistic GLMM is necessarily better in this situation than a paired t-test as done by the original authors. Which analysis approach one ultimately prefers depends on which assumptions one thinks is more believable. For the binomial GLMM, one assumes that participants are normally distributed around their condition means in logistic space. For the t-test, one essentially assumes that participants are normally distributed around their condition means in accuracy space. Which assumption is correct is not an easy question. Recently, I am becoming more convinced by analyses that do not require a data transformation (such as to logistic space) for accuracy data. One reason for this is a series of papers (e.g., Gomilla, 2021; Jaccard & Brinberg, 2021) arguing for the usage of models on a non-transformed probability space. In this context, models such as paired t-test or RM-ANOVA for accuracy data is also known as the linear probability model.”
We appreciate this perspective. From our reading of the papers cited by the reviewer, they argue that in many practical cases, the choice between a linear probability model and a transformed (e.g., logistic) model may not substantially affect results. In such situations, the argument for preferring the linear probability model is primarily one of parsimony: in many cases, the models yield equivalent conclusions, we should opt for the simpler of the two models. However, in our case we clearly show that this choice does make a difference. As mentioned before, we believe that this derives from how the two approaches treat performance changes near ceiling and believe that the GLMM approach is more faithful to the underlying data-generating process.
As an additional comment, Gomila (2021) presents simulation results in which the proportion of a binary outcome in a control group is compared to that in a treatment group; the take-home message is that linear and logistic regressions yield similar regression weights (the average treatment effect). In these simulations (Figure 1), the treatment always caused, on average, an increase of 0.08 (i.e. 8 percentage points). The presented simulations thereby avoid boundary conditions on the averages, because the highest control proportion used is 0.9 (which means that the linear model can still provide an unbiased estimate of the average change up to 0.98). Gomila’s demonstration by simulation therefore does not cover the empirical case we present here.
We also agree with the reviewer that the appropriateness of an analytical approach depends on the assumptions one finds most plausible. However, we would like to point out that the assumption that participants’ latent means and slopes are normally distributed is an assumption underlying a frequentist logistic GLMM approach, but not necessarily of Bayesian models. In a Bayesian model estimated via MCMC, assigning a Normal prior to random effects does not force their posterior distributions to be Normal: if the data suggest skew or other deviations, the posterior will reflect that. This is because each participant-level mean effect has its own posterior distribution (i.e. is a parameter to be estimated). This differs from frequentist implementations (e.g., using lme4 with REML), where the Normal random-effects assumption is rigidly imposed through the estimation of the variance of a Normal distribution, with each participant-level offset being then determined by the BLUP “Best Linear Unbiased Predictors” (see e.g. Baayen et al., 2008).
We thank both reviewers once again for their careful reading and insightful comments on our paper. We believe the updated manuscript is now clearer and improved relative to the original submission.
Best Wishes,
Thomas Wallis & Joshua Martin
References
Baayen, R. H., Davidson, D. J., & Bates, D. M. (2008). Mixed-effects modeling with crossed random effects for subjects and items. Journal of Memory and Language, 59(4), 390–412. https://doi.org/10.1016/j.jml.2007.12.005
Gomila, R. (2021). Logistic or linear? Estimating causal effects of experimental treatments on binary outcomes using regression analysis. Journal of Experimental Psychology: General, 150(4), 700.
Jaccard, J., & Brinberg, M. (2021). Monte Carlo simulations using extant data to mimic populations: Applications to the modified linear probability model and logistic regression. Psychological Methods, 26(4), 450.

---

## Round 2 · List of Changes

The main point of both reviewers relates to the rationale for our approach and what could explain the divergent findings of the two methods. Reviewer 1 writes that “The rationale for the different analysis approach is unclear” and “The authors should clarify what is the main reason for why this new analysis results in opposite conclusion compared to the original paper” while reviewer 2 writes that “It is unclear why the Bayesian mixed-effects analysis should be preferred to the one reported in the original paper” and relatedly “What actually is wrong with the original author’s approach (t-test)?”
Response:
We agree that this could have been made clearer in the original manuscript. Our original motivation for the reanalysis derived from our observation of a ceiling effect in the control data, where many data-points are compressed together (Figure 1A). T-tests on difference scores underestimate these changes by treating percentage differences identically across the performance bound. A t-test implicitly assumes a linear model with Normal-distributed errors, which as a generative model for accuracy data is impossible (since it can predict performance greater than 100%). A logistic regression model, on the other hand, represents performance changes on an unbounded log-odds scale, which can mitigate this issue by being more sensitive to changes occurring near ceiling. Additionally, logistic regression typically assumes a binomial error distribution, which is appropriate for data in which the unit of observation is a binary outcome. We have now added the following to the introduction to make the rationale for this approach clearer:
“However, because accuracy is bounded and some control participants perform at or near ceiling, t-tests on difference scores may underestimate the improvement in controls by treating a 5% change near 95% as statistically equivalent to one at 50% (i.e. assuming that probability is linear and variance is Normally-distributed). We therefore examined whether this effect is robust when analyzing trial-level accuracy using a logistic regression model [2], which represents performance changes on the unbounded log-odds scale (Figure 1B).”
Furthermore, we believe that the observed differences in conclusions arise from this fundamental distinction between how the two methods treat performance changes on bounded versus unbounded scales. We have added this to the conclusion as an interpretation of the divergent findings:
“Expressing performance changes as raw percentages (assuming a linear probability model) results in the color and greyscale conditions in the control group being compressed closer together (Figure 1A). This compression directly creates the interaction effect observed using t-tests. We argue that assuming a linear probability model with Normal measurement error is therefore inappropriate here. Using a generative model that treats this performance data on its natural scale (log odds and binomial errors) mitigates the effects of performance saturation by attributing greater weight to changes at or near ceiling.”
Four smaller points raised by reviewer 2
In addition to the main point above, there are four smaller points that we would like to address in turn. We list the quote relevant to each point and our brief response underneath.
1. “Taken together, in a revision the authors need to make absolutely clear that the evidence for the interaction hinges on whether one is willing to believe the linear probability model (as done by Vogelsang et al., 2024) or the binomial GLMM with logistic link function.”
We agree that ultimately the conclusion that one draws from the study hinges on which model one thinks is more faithful to the underlying generative process. We have added the following as a concluding sentence to reflect this:
"Therefore, the conclusion that late-sighted individuals rely more on color information than individuals with normal visual development from these data appears to hinge on which modeling approach one views as most appropriate for the underlying data-generating process."
2. “Code for Figure 1 in report not included in provided R code.”
We thank the reviewer for pointing this out. We have now updated the repository to include the code for generating the figure to make sure every aspect of the report is transparent and reproducible.
3. “The conclusion talks about two different criticisms: The differences in results between t-test and Bayesian-GLMM and learning as a "competing causal explanation". However, the paper only focuses on the differences in analysis so the second issue does not follow from the presented analysis.”
We agree with the reviewer that this statement does not directly derive from our alternative statistical analysis. While we feel it is an important design limitation related to the experiment, we have decided to remove this point as it is not directly relevant to the report at hand.
4. “To be transparent, I do not think that using a binomial-logistic GLMM is necessarily better in this situation than a paired t-test as done by the original authors. Which analysis approach one ultimately prefers depends on which assumptions one thinks is more believable. For the binomial GLMM, one assumes that participants are normally distributed around their condition means in logistic space. For the t-test, one essentially assumes that participants are normally distributed around their condition means in accuracy space. Which assumption is correct is not an easy question. Recently, I am becoming more convinced by analyses that do not require a data transformation (such as to logistic space) for accuracy data. One reason for this is a series of papers (e.g., Gomilla, 2021; Jaccard & Brinberg, 2021) arguing for the usage of models on a non-transformed probability space. In this context, models such as paired t-test or RM-ANOVA for accuracy data is also known as the linear probability model.”
We appreciate this perspective. From our reading of the papers cited by the reviewer, they argue that in many practical cases, the choice between a linear probability model and a transformed (e.g., logistic) model may not substantially affect results. In such situations, the argument for preferring the linear probability model is primarily one of parsimony: in many cases, the models yield equivalent conclusions, we should opt for the simpler of the two models. However, in our case we clearly show that this choice does make a difference. As mentioned before, we believe that this derives from how the two approaches treat performance changes near ceiling and believe that the GLMM approach is more faithful to the underlying data-generating process.
As an additional comment, Gomila (2021) presents simulation results in which the proportion of a binary outcome in a control group is compared to that in a treatment group; the take-home message is that linear and logistic regressions yield similar regression weights (the average treatment effect). In these simulations (Figure 1), the treatment always caused, on average, an increase of 0.08 (i.e. 8 percentage points). The presented simulations thereby avoid boundary conditions on the averages, because the highest control proportion used is 0.9 (which means that the linear model can still provide an unbiased estimate of the average change up to 0.98). Gomila’s demonstration by simulation therefore does not cover the empirical case we present here.
We also agree with the reviewer that the appropriateness of an analytical approach depends on the assumptions one finds most plausible. However, we would like to point out that the assumption that participants’ latent means and slopes are normally distributed is an assumption underlying a frequentist logistic GLMM approach, but not necessarily of Bayesian models. In a Bayesian model estimated via MCMC, assigning a Normal prior to random effects does not force their posterior distributions to be Normal: if the data suggest skew or other deviations, the posterior will reflect that. This is because each participant-level mean effect has its own posterior distribution (i.e. is a parameter to be estimated). This differs from frequentist implementations (e.g., using lme4 with REML), where the Normal random-effects assumption is rigidly imposed through the estimation of the variance of a Normal distribution, with each participant-level offset being then determined by the BLUP “Best Linear Unbiased Predictors” (see e.g. Baayen et al., 2008).
We thank both reviewers once again for their careful reading and insightful comments on our paper. We believe the updated manuscript is now clearer and improved relative to the original submission.
Best Wishes,
Thomas Wallis & Joshua Martin
References
Baayen, R. H., Davidson, D. J., & Bates, D. M. (2008). Mixed-effects modeling with crossed random effects for subjects and items. Journal of Memory and Language, 59(4), 390–412. https://doi.org/10.1016/j.jml.2007.12.005
Gomila, R. (2021). Logistic or linear? Estimating causal effects of experimental treatments on binary outcomes using regression analysis. Journal of Experimental Psychology: General, 150(4), 700.
Jaccard, J., & Brinberg, M. (2021). Monte Carlo simulations using extant data to mimic populations: Applications to the modified linear probability model and logistic regression. Psychological Methods, 26(4), 450.

---

## Editorial Decision

editorial_decision: